

# ProcGCN: detecting malicious process in memory based on DGCNN

Heyu Zhang, Binglong Li, Shilong Yu, Chaowen Chang, Jinhui Li and Bohao Yang

College of Cryptographic Engineering, Information Engineering University, Zhengzhou, Henan, China

## ABSTRACT

The combination of memory forensics and deep learning for malware detection has achieved certain progress, but most existing methods convert process dump to images for classification, which is still based on process byte feature classification. After the malware is loaded into memory, the original byte features will change. Compared with byte features, function call features can represent the behaviors of malware more robustly. Therefore, this article proposes the ProcGCN model, a deep learning model based on DGCNN (Deep Graph Convolutional Neural Network), to detect malicious processes in memory images. First, the process dump is extracted from the whole system memory image; then, the Function Call Graph (FCG) of the process is extracted, and feature vectors for the function node in the FCG are generated based on the word bag model; finally, the FCG is input to the ProcGCN model for classification and detection. Using a public dataset for experiments, the ProcGCN model achieved an accuracy of 98.44% and an $F1$ score of 0.9828. It shows a better result than the existing deep learning methods based on static features, and its detection speed is faster, which demonstrates the effectiveness of the method based on function call features and graph representation learning in memory forensics.

## INTRODUCTION

With the development of computer and Internet technology, malware has progressed rapidly. According to AV-TEST statistics, more than 70 million new Windows malware issues emerged in 2022, posing a serious threat to cyberspace security (*AV-TEST, 2023*). Therefore, it is necessary to develop automatic malware detection methods.

A typical memory forensic scenario is shown in the Fig. 1. The user can be infected by visiting malicious websites or executing malicious email attachments. Then, the malware bypasses antivirus software, launch malicious processes in memory, and deletes itself from the disk. When the victim system is found to be infected, it is always too late to catch the original malicious samples from the disk. Therefore, memory forensics technology is used to obtain a complete memory image, extract malicious processes and threads, and analyze attack metrics. This article aims to detect malicious processes from complete memory images in the memory forensic scenario.

Corresponding author
Binglong Li, lbl2017@163.com

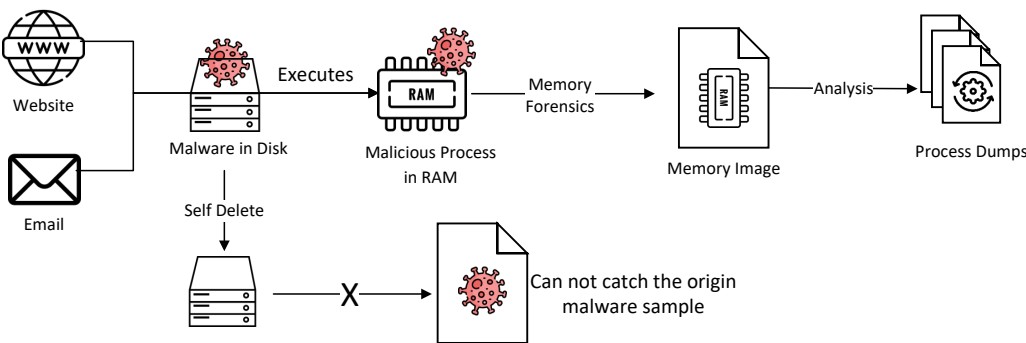

**Figure 1** **The memory forensics scenario.** Image source credits: Website icons created by Freepik–Flaticon, Email icons created by Freepik–Flaticon, Server icons created by Freepik–Flaticon, Ram icons created by Freepik–Flaticon, Process icons created by Freepik–Flaticon, Virus icons created by Freepik–Flaticon.

At present, there are mainly three types of malware detection methods, namely, static analysis, dynamic analysis, and memory-based analysis methods. Static analysis methods use static features such as string, opcode, API call, byte sequence, static control flow graph (CFG), *etc* (*Galloro et al., 2022*). Dynamic analysis methods execute malware samples in a controllable environment and monitor their behavior (*Or-Meir et al., 2019*). Generally, sequential features are used to judge malice in dynamic analysis. However, in the scenario shown in Fig. 1, there are no original executable malware samples and no complete attack sequences, so the dynamic analysis method is not applicable for memory forensics. The memory-based analysis is an effective malware detection method, which has attracted more and more attention in recent years. Although malware can be hidden by encryption and packaging, during the execution process, all processes will occupy key codes and data segments in memory for running, which usually represent the behavior features of malware (*Dai et al., 2018*).

In recent years, some studies have been conducted on combining deep learning technology with memory forensics technology for malware detection. For example, *O'Shaughnessy & Sheridan (2022)* proposed to convert process memory dump into images mapped by space-filling curves and extract visual features for malware detection; *Dai et al. (2018)* dumped and converted processes into gray images and classify them with multilayer perceptron to detect malicious software; *Bozkir et al. (2021)* transformed process dump into a color image and then extracted visual features to classify and detect malicious processes. Most of these methods convert the original bytes of malicious processes into images for classification, but *Babaagba & Adesanya (2019)* and *Demetrio et al. (2019)* proposed that the byte features of malicious software are easily affected by mutation and confusion methods, thus declining byte-based detection accuracy. *Jiang, Turki & Wang (2018)* pointed out that API call features can represent the functions of malware more robustly than byte features. They used function call graph (FCG) to represent the features of the software, abstracted malware classification as a graph classification task, used Node2Vec to embed features, and then input them into an automatic encoder for

classification. However, Node2Vec model can not aggregate node features in directed graph, which limits the performance of this study.

The graph neural network (GNN) is a new graph representation learning method and has achieved good results in graph classification tasks. For example, *Zhang et al. (2018)* proposed an end-to-end graph neural network architecture called DGCNN, which can directly read graphs and embed graphs and has good performance. Therefore, researchers use GNN to embed graphs into feature vectors to solve the problem of malware detection or classification. For instance, *Zhang & Li (2020)* adopted GCN (Graph Convolutional Network) to classify Android applications to detect malware, which has achieved good results. However, we noticed that GCN with excellent performance is hardly used in Windows malware detection. Moreover, existing research almost only focuses on system APIs without considering the impact of general functions on program behavior.

This article combines GNN with memory forensics technology and proposes a malicious process detection method called ProcGCN (malicious process detection based on Deep Graph Convolutional Neural Network). This method extracts the process dump from the whole system memory image, generates the FCG of the process, and converts each function node in the FCG into a feature vector according to the function name. Then, the FCG is input to the DGCNN model to generate the feature vector of the graph, and finally, the feature vector of the graph is input to the fully connected network to classify and detect malware. The proposed method is based on memory image analysis and can resist obfuscated or encrypted malware attacks. Meanwhile, the GCN method is introduced into memory forensics, and the relationship between the internal functions in the process can be aggregated. Experimental results indicate that this method achieves an accuracy of 98.44% and an F1 score of 0.9828.

The rest of this article is organized as follows. 'Related Works' introduces the related work. 'ProcGCN' presents the proposed malicious process detection method based on GNN. In 'Experiment and Discussion', the experimental results, discussion, and analysis of the algorithm are introduced. Finally, this article is summarized.

## RELATED WORKS

This section mainly discusses malware analysis methods based on function call features, graph representation learning, and memory forensics.

The malware detection methods based on dynamic API call sequences have been widely studied. For example, *Agrawal et al. (2018)* input the $n$-gram of API names and string parameters into a stacked LSTM model for malware detection. *Li et al. (2022)* represented the API call sequence and parameters of malware with a directed graph and classify them with the GCN model. However, these methods are not suitable for detecting malicious processes in memory images because the API call sequences cannot be restored.

In recent years, malware detection methods based on static API call features mainly express the function call relationship of software in the form of graphs, which are often combined with graph representation learning methods. For example, *Bai, Shi & Mu (2019)* proposed to take FCG as the signature of programs, and they adopted two graph

isomorphism algorithms to identify known malware and its variants. *Jiang, Turki & Wang (2018)* developed a method called DLGraph, which uses FCG to represent software features, uses Node2Vec to embed features, and then inputs them into an automatic coding machine for classification. These studies demonstrate the effectiveness of applying the FCG to malware detection. However, Node2Vec model can not aggregate node features in directed graph, which limits the performance of DLGraph.

GNN is widely used to detect Android malicious applications. For example, *Yang et al. (2021)* proposed an Android malware detection method called DGCNDroid, which inputs a function call subgraph containing sensitive APIs into DGCNN to detect and classify malicious applications. *Zhang & Li (2020)* proposed a method to extract semantic structure features of Android application codes and used GNN to classify the extracted code semantic graphs. These studies indicate that the GNN model can effectively classify software functional features based on graph representation.

At present, most malware detection technologies based on memory forensics are based on static byte features. For example, *O'Shaughnessy & Sheridan (2022)* presented a hybrid framework for malware classification, which converts static malware executable files and dynamic process memory dumps into images mapped by space-filling curves and extracts visual features from them for classification; *Dai et al. (2018)* proposed to extract process dump from memory, convert it into a gray image, and classify and detect malicious software with multi-layer perceptron; *Bozkir et al. (2021)* put forward a method of dumping and converting processes into color images and then using machine learning method to classify and detect malicious processes based on GIST and gradient histogram features. *Li et al. (2021)* proposed a malicious code fragment forensics algorithm based on a deep fully connected network, which embeds fixed-length malicious code fragment bytes into feature vectors and inputs them into fully connected networks for classification; *Khalid et al. (2023)* proposed to transform the state information such as process, service list, callback, and registry in the memory image into feature vectors and then use the classical machine learning algorithm to classify and detect file-less malware. Different from the above methods, this article uses FCG to represent the functional features of malware and combines GNN to detect and classify malicious processes in a memory image.

Additionally, datasets are crucial for research on machine learning. However, the datasets in the field of malware detection is very limited, such as the Microsoft Malware Classification Challenge (BIG2015) dataset (*Ronen et al., 2018*), the "Malimg" dataset (*Nataraj et al., 2011*), the "MaleVis" dataset (*Bozkir, Cankaya & Aydos, 2019*), the "Dumpware10" dataset (*Bozkir et al., 2021*), the PE file dataset published by *Fang et al. (2020)*, and the Win10 memory snapshot dataset published by *Sadek et al. (2019)*. Among them, the Microsoft BIG2015 dataset removes the file headers of all PE files in the dataset for security reasons, thus making them non-executable. Malimg, MaleVis, and Dumpware10 datasets only contain image files after conversion and cannot restore the original executable files, so they are only suitable for malware detection methods based on image visual features. The dataset published by Sadek et al. are memory snapshots captured after running Metaspolit and packing tools in the Win10 operating system, which is only suitable for studying penetration and obfuscation attack. Only the dataset published by *Fang et al. (2020)* contains benign

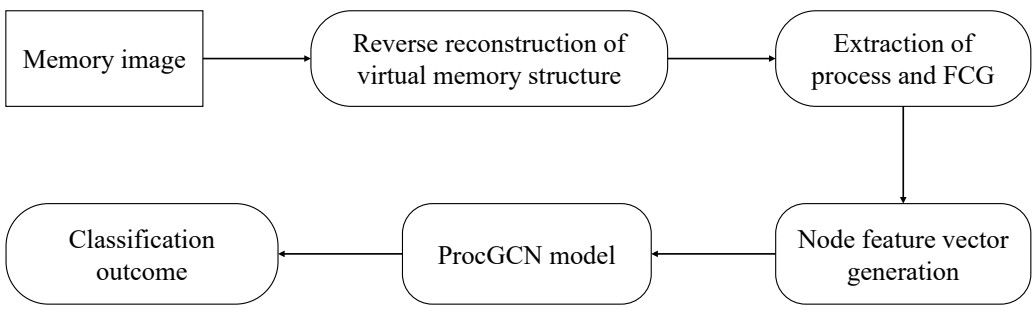

**Figure 2  The GCN-based malicious process detection framework.**

and malicious software and can run again in experimental environment. This dataset is used in this article after preprocessing.

## PROCGCN

To detect malicious processes in memory images, this study proposes a malicious process detection method based on the DGCNN called ProcGCN, and a framework of malicious process detection. The main workflow of the framework is shown in Fig. 2. Firstly, the virtual memory space of the memory image is reconstructed, and the memory dump of each process is extracted. Then, the static FCG is established from the process dump, the functional semantics of the process are represented by FCG, and the malicious process detection is abstracted as a binary classification task of the graph. Subsequently, the feature vectors are generated for each function node in the FCG, and the FCG is input into the ProcGCN model to obtain the benign or malicious classification results of the process FCG.

### Reverse reconstruction of virtual memory space

Modern operating systems generally use the virtual memory and Address Space Layout Randomization (ASLR) mechanism to manage memory space, so the physical memory pages in memory image files are logically discontinuous. Only by completing the reverse reconstruction of the virtual memory space of the memory image can the processes in the memory image be identified and extracted.

The mapping relationship from the virtual memory page to the corresponding physical memory page is established by the page table. According to the virtual memory and physical memory conversion method given by *Russinovich, Solomon & Ionescu (2012)*, once the Directory Table Base (DTB) address of a process is found, the virtual memory space of the process can be reconstructed. The steps of the reverse reconstruction of virtual memory space are as follows:

(1) The current memory forensics framework provides configuration files of the memory management structure and global variables in different versions of operating systems. The KPCR and _KDDEBUGGER_DATA64 structures and signature strings can be read

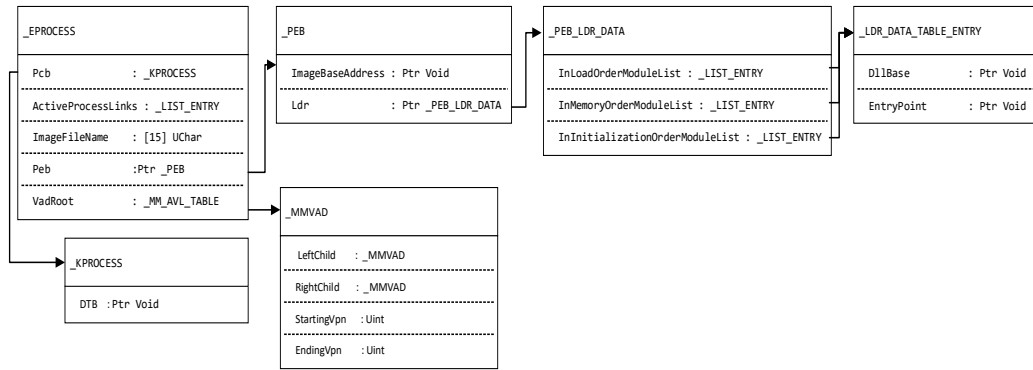

**Figure 3  The structure of Windows system process management.**

from the configuration. Then the signatures are scanned in the memory image, and the _KDDEBUGGER_DATA64 and KPCR structure are located and reconstructed.

(2) Read the kernel DTB from the KPCR structure and reconstruct the virtual memory space of the system kernel space.

(3) Read the active process linked list address named PsActiveProcessHead from the _KDDEBUGGER_DATA64 structure, which points to the virtual address of the EPROCESS of the first process in the kernel region, and rebuild the EPROCESS structure of the process according to the definition of the EPROCESS structure in the configuration file (as shown in Fig. 3).

(4) Read the process's DTB from EPROCESS.Pcb.DTB and rebuild the process's virtual memory space.

(5) EPROCESS.ActiveProcessLinks traverses the EPROCESS structure of all processes, obtains the EPROCESS.Pcb.DTB of each process, and completes the reconstruction of the virtual memory space of all processes.

So far, the kernel and virtual memory space of all processes can be accessed, and the reverse reconstruction of virtual memory space is completed.

## Extraction of process function call graph

To use FCG to represent the functional features of a process, it is necessary to extract the process dump, find the entry point function address of the process dump through the process information in memory, disassemble the memory dump completely from the entry point function, find out all the function call instructions and their addresses in the process, and find the function import address table (IAT) to determine the external functions, and generate FCG according to the function call relationship.

Current mainstream disassembly tools, such as IDA_PRO (*Hex Rays*), support efficient and stable disassembly and can extract the PE executable file FCG, so this article converts the process dump to a PE executable file and adopt the disassembly tool to generate FCG. The specific process is as follows:

(1) Read the process base address from EPROCESS.Peb.ImageBaseAddress; Obtain the virtual address descriptor (VAD) root node address from EPROCESS.VadRoot and reconstruct the VAD tree structure (as shown in Fig. 3).

(2) Find the VAD node where the base address of the process is located, read the VAD.EndingVpn of the node, and obtain the end address of the VAD block. The range from the base address to the end address of the process is the address range that the process needs to dump.

(3) The first pointer to EPROCESS.Peb.Ldr.InMemoryOrderModuleList points to the process's _LDR_DATA_TABLE_ENTRY structure and reads EntryPoint to obtain the entry point function.

(4) To reorganize the exported process dump into a PE format file, the virtual address (VA) and the relative virtual address (RVA) need to be re-aligned according to the section offset; then, the section table of the PE header is filled according to the reconstructed offset, the import function table, the entry point function address obtained in step (3), and other characteristic bytes fixed in the PE header. In this way, the reorganized PE file is obtained.

At this time, the process dump extraction and PE format reconstruction are completed. However, the reconstructed PE file is usually larger than the original PE file and contains some initialized and modified variables and other runtime data. For instance, some fields in the .data/.bss section may be changed or assigned values, and even partially packaged malware will decrypt and decompress the original instruction data, making the reconstructed PE file much different from the original file.

Although the restructured PE file may not execute properly, it can still be statically analyzed by disassembly tools. The restructured PE file is input into the disassembly tool to generate FCG. FCG is a directed graph, where each node represents a function, and the directed edge in the graph represents the calling relationship between function nodes (as shown in Fig. 4). A large number of local functions in the form of "sub_xxxxxx" do not contain functional semantics, but their function can be reflected through the calling relationship of their neighbor nodes. Therefore, the FCG can represent the functional semantics of the process.

## Generation of function node feature vector

To classify FCG using a deep learning model, it is necessary to extract the node vector of FCG. The name of a function node generally reflects the functional semantics of the function, so the feature vector of the node is generated based on the function name.

As shown in Fig. 4, the types of function nodes in an FCG include entry point functions, local functions, external functions, and library functions, among which local functions are named by the software author, and the name may contain available functional semantic information, while another part of the local functions in the form of "sub_xxxxxx" are only related to addresses and do not contain functional semantics; External functions are functions imported into a program from external files, and they can be divided into system API functions and general external functions. The names of system API functions contain specific functional semantics and the same functional semantic information in different programs, while the names of general external functions do not necessarily contain available

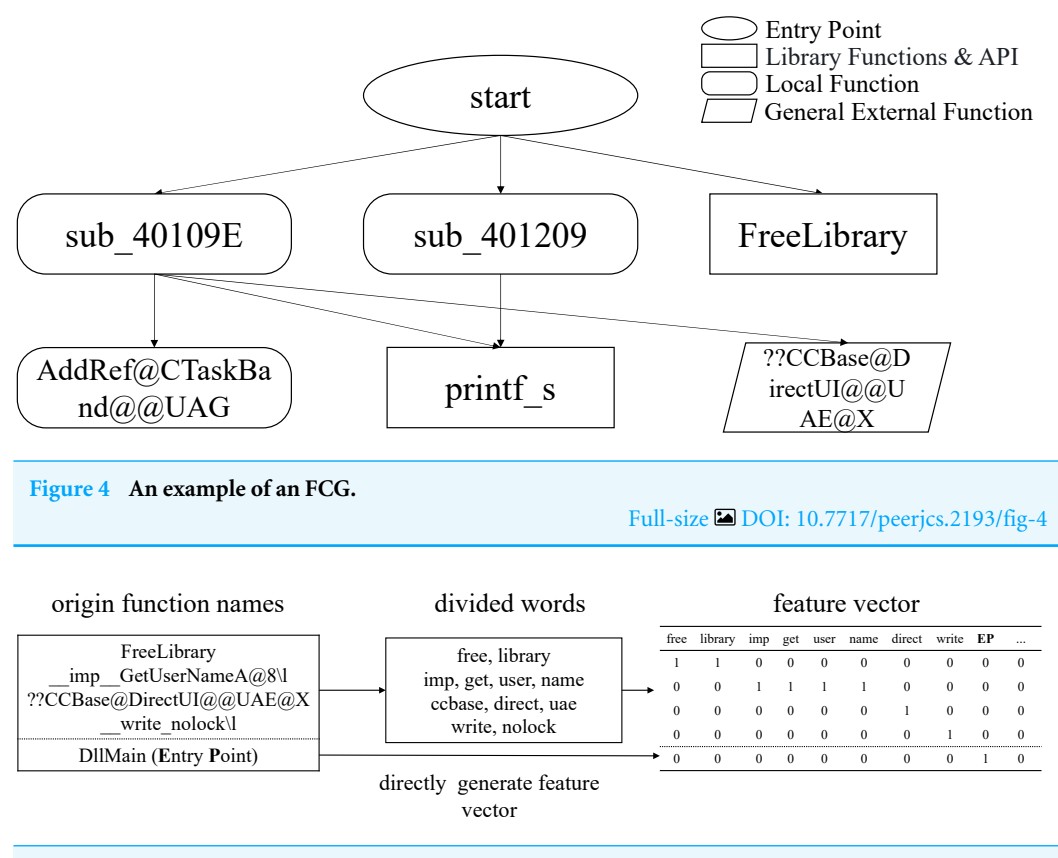

**Figure 4**  An example of an FCG.

**Figure 5**  An example process of generating feature vectors from function names.

functional semantic information; Library functions are provided by C/C + + standard. The names of these functions also contain general functional semantics, but the names of library functions may vary slightly from compiler to compiler.

Except for internal functions, the names of other function nodes all contain functional semantic information, so the feature vector of this node can be generated according to the function name based on the word bag model. The general naming method of function names is Camel-Case or Underscore-Case. Some function names also contain information about the function library or class, and they can be divided by other special symbols. If encode directly by name, the feature vector space will be too large, so this article divides the function name into vocabulary and extract semantic information as much as possible. The specific method is described below (as shown in Fig. 5):

(1) Collect all external function names and library function names in the dataset.

(2) Divide names into words by special characters and uppercase letters and convert them all to lowercase while discarding numbers and words less than 2 characters.

(3) Take part of the words that appear most frequently to form a vocabulary and delete the "unknown" in it to prevent the confusion of functional features caused by unknown functions; Since the entry point function has unique and definite functional characteristics,

an additional feature dimension "EP" (Entry Point) is generated based on the vocabulary to mark the entry point function.

(4) Re-traverse the function nodes, divide the function name into words according to step (2), and generate feature vectors according to the vocabulary. As shown in Fig. 5, the words in the name are marked as 1 in the corresponding dimension; "ccbase", "uae", and "nolock" are not in the vocabulary and are discarded directly; The entry point function node directly generates the feature vectors according to "EP" without considering the original function name.

## ProcGCN model

The ProcGCN model consists of a DGCNN graph convolution model and a fully connected network classifier. DGCNN is an efficient end-to-end model for whole graph classification model. This article builds ProcGCN based on DGCNN. The hyper-parameters and the number of layers of the full connection model in ProcGCN are obtained by grid search, and the search range is listed in Table 1. The model structure obtained from the best search results is presented in Fig. 6.

The ProcGCN model is an end-to-end model, which takes an FCG with node feature vectors as input directly. In the DGCNN module, the feature vectors of its neighbors are aggregated for each node through a four-layer GCN. For FCG, local functions without functional semantic features can aggregate the features of their neighbors with features through graph convolution, thus reflecting their unique calling relationship features and realizing feature differentiation from other local function nodes. Specifically, as in "sub_40109e" and "sub_401209" in Fig. 4, the names of the two local functions have no functional semantics, so the initial feature vectors are $[0...0]$, but the functions called by these two functions are different, _i.e.,_ neighbor nodes with different features. After graph convolution, the features of different neighbor nodes are aggregated, resulting in different feature differences and enriching the whole FCG graph.

The SortPooling layer connects and sorts the vertices after the graph convolution layer, enables the model to remember the input order during backpropagation, and then realizes parameter learning. Meanwhile, the arbitrary input graph structure is changed into a fixed size and input to the CNN layer. Then, the feature of the graph is further extracted by 2 layers of CNN, and the feature vector of the whole graph node is aggregated. After dimensionality reduction by MaxPool, the classifier of two layers of FC (full connection layer) is input, and finally, the malicious probability of FCG is output.

## EXPERIMENT AND DISCUSSION

### Dataset

The dataset collected by _Fang et al. (2020)_ was used in the experiment, which included 3,628 malicious samples collected from VirusShare and 3,746 benign samples collected by major software download platforms.

(1) To ensure the accuracy of positive and negative sample labels, this study filtered out EXE files from the original dataset, obtain the report of each PE file on VirusTotal, added

Table 1  The setting of hyperparameters for ProcGCN.

| Hyperparameter | Range | Best value |
|---|---|---|
| K Param of DGCNN | 35, 45, 60 | 35 |
| DGCNN layers | 3, 4, 5 | 4 |
| DGCNN layer size | 16, 32, 64 | 32 |
| Hidden layers | 1, 2, 3 | 2 |
| Hidden layer1 size | 32, 64, 128 | 128 |
| Hidden layer2 size | 32, 64, 128 | 64 |
| Learning rate | $1 \times 10^{-4}, 5 \times 10^{-4}, 1 \times 10^{-3}, 5 \times 10^{-3}$ | $1 \times 10^{-3}$ |

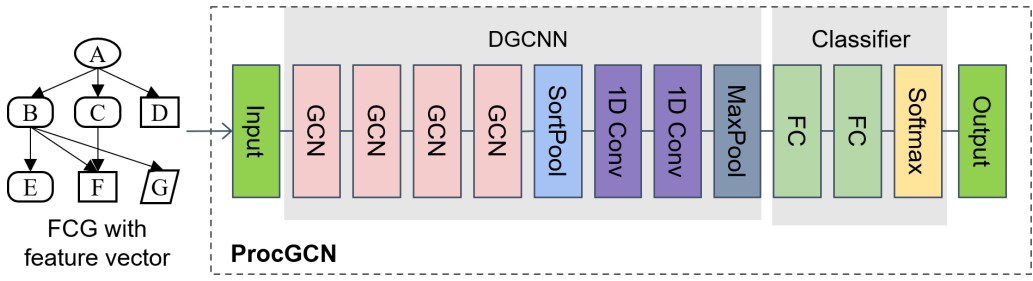

**Figure 6**  The schematic diagram of the ProcGCN model.

samples of none malicious reports to the benign software set, and added samples of more than 30 malicious reports to the malicious software set.

(2) The Cuckoo sandbox and VirtualBox virtual machine were used to build a Windows 7 virtual environment with 1 GB memory. The EXE files are loaded using cuckoo's default settings.

(3) According to the logs generated by Cuckoo, the PE format dump of all processes in the process tree of sample process in the memory image was extracted by using the "procdump" plugin of Volatility (*The Volatility Foundation*), and the FCG of the process was extracted by using IDA_Pro. All processes generated by benign and malicious samples were marked as benign and malicious, respectively.

(4) Finally, 2,348 benign and 2,786 malicious FCG were obtained, 5,134 samples in total. The complete dataset is divided into the training set and the test set at a ratio of 8:2. The number distribution of nodes and edges in the dataset is shown in Figs. 7 and 8, the malware categories are shown in Fig. 9, and the number distribution of four types of functions is shown in Table 2.

## Experimental environment and procedures

This experiment was performed on a PC with Intel i7-13700K CPU, Nvidia RTX 3060 12GB GPU, and 32GB RAM. The ProcGCN model was implemented with PyG (PyTorch Geometric) (*Fey & Lenssen, 2019*) on top of the PyTorch platform. The experimental steps are as follows:

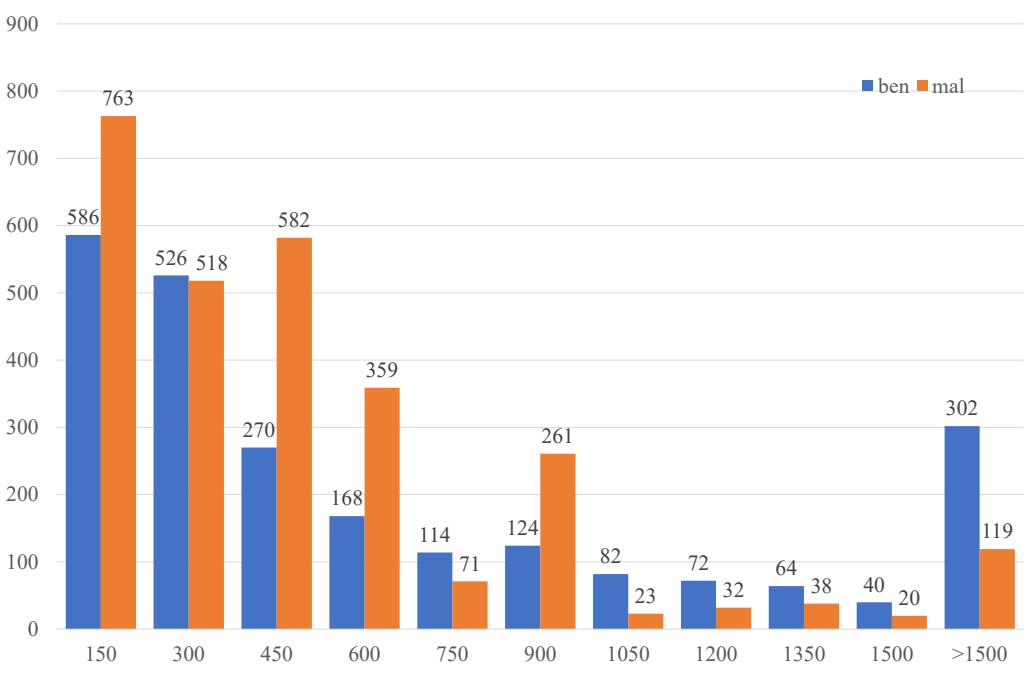

**Figure 7** The histogram of sample function (node) number distribution.

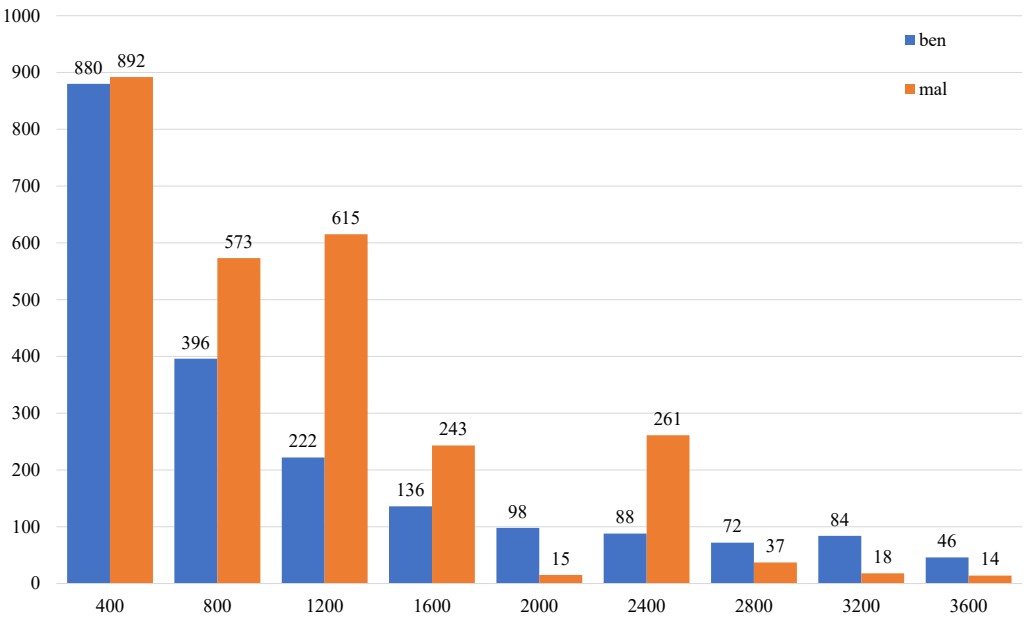

**Figure 8** The histogram of sample call relationship (edge) number distribution.

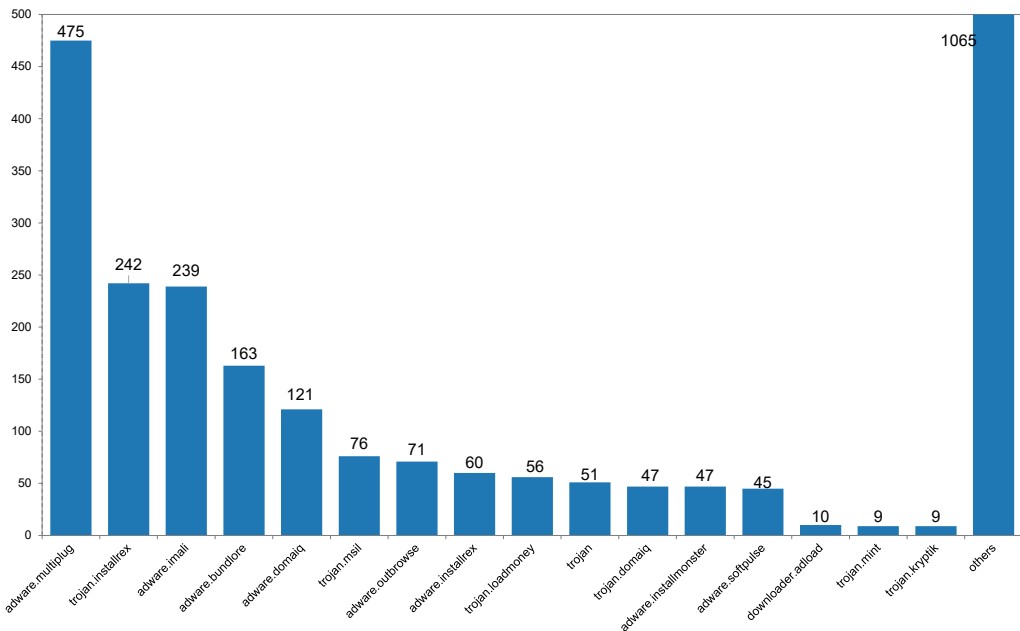

**Figure 9  The histogram of malware sample category distribution.**

(1) Establish a dataset as introduced in 'Dataset'. The FCG is generated by IDA_Pro. The four types of nodes are distinguished by colors, namely the entry point function, local function, external function, and library function. Then, use Networkx (*Hagberg, Schult & Swart, 2008*) to read the FCG files in the dot format.

(2) Construct a vocabulary. For all FCG in the dataset, read the function name of each node and divide the words following the method in 'ProcGCN', take out 1,443 words in the first 1,000 frequency, delete the "unknown", and add the feature "EP" representing the entry point function node. In this way, the vocabulary length is 1,443.

(3) Generate feature vectors of nodes. Since the FCG generated by IDA_Pro marks the entry point function node with colors, the function name of the entry point does not need to be considered, the entry point function is directly recognized, and the feature vector is generated according to the color; for other nodes, feature vectors are generated by name according to the method in 'ProcGCN'.

(4) Construct and train the ProcGCN model. The layers and hyper-parameters of the ProcGCN model are obtained by grid search, and the search range is shown in Table 1. The model structure obtained from the best search results is shown in Fig. 6.

To demonstrate the advantages of ProcGCN in memory forensics scenarios, ProcGCN is compared with Malconv (*Raff et al., 2018*) and Bozkir's method (*Bozkir et al., 2021*). The ProcGCN method is not compared with dynamic detection methods such as DMalNet (*Li et al., 2022*) because the complete sequence of API calls and their parameters cannot be extracted from memory images. Malconv is an end-to-end model based on static byte sequence. Due to the high memory usage of this model, the maximum length of the input byte sequence is limited to 200,000 in this experiment. Bozkir's SMO method is a memory

**Table 2  The number distribution of four types of functions.**

|  | Malicious | Benign | Total |
|---|---|---|---|
| sub_xxx (unknown local func) | 1,049,118 (12.8%) | 1,055,544 (12.9%) | 2,104,662 (25.7%) |
| Meaningful local func | 74,193 (0.9%) | 1,011,628 (12.4%) | 1,085,821 (13.3%) |
| External func | 303,261 (3.7%) | 3,938,598 (48.2%) | 4,241,859 (51.9%) |
| Library func | 504,810 (6.2%) | 238,578 (2.9%) | 743,388 (9.1%) |
| Total | 1,931,382 (23.6%) | 6,244,348 (76.4%) | 8,175,730 |

forensic analysis method that converts the binary dump of processes into RGB images, and then uses SMO algorithm with the Gaussian kernel based on the GIST feature for classification. The experimental results are as follows.

## Experimental results

As shown in Table 3, ProcGCN achieved an accuracy of 98.44%, a recall rate of 98.69%, an F1 score of 0.9828, and the false alarm rate (FPR) of 0.0199 on the test set. ProcGCN achieved the best metrics among the three methods.

For time and memory consumption, as shown in Table 4, ProcGCN preprocessing process consists of decompiling, extracting FCGs, and feature engineering, which consumes the most time. The SMO method encodes the samples into images and extracts their visual features, which consumes less time. In this experiment, the preprocessing time of the Malconv model includes serialization and disk IO time. The Malconv model requires the minimum preprocessing time. In terms of prediction speed, the SMO algorithm is a machine learning algorithm with the fastest prediction speed. Compared with Malconv, the ProcGCN model has significant advantages in prediction speed.

## Analysis and discussion

In terms of prediction results, the ProcGCN model achieved the best accuracy, recall, F1 score, and FPR value among the three methods in experiments. The experimental results show the superiority of the process FCG in characterizing process functions and the effectiveness of GNN in detecting malicious processes.

Malconv is an end-to-end model designed to detect the original PE based on the file byte sequence and is not suitable for memory forensics. The memory process adds a large number of padding bytes to align the memory paging, leading to decreasing information density. However, After the input sequence truncation, the byte features may be reduced, which will affect the accuracy of Malconv.

The SMO method proposed by *Bozkir et al. (2021)* converts bytes based on memory dump into images and extracts visual features for classification. The byte length of the samples in the dataset used is quite different, and in this case, the byte length affects the visual features of the converted images, decreasing the accuracy of the SMO algorithm. In contrast, ProcGCN uses FCG features and is not easily affected by sample byte length.

In terms of prediction speed, the ProcGCN model performs better than Malconv but worse than the SMO algorithm. This is because the parameter quantity of the ProcGCN model is 235,082, which is far less than 1,034,625 of Malconv, but its computation amount

**Table 3 The evaluation metrics.**

|  | Accuracy | FPR | Precision | Recall | *F*1-score |
|---|---|---|---|---|---|
| ProcGCN | 0.9844 | 0.0199 | 0.9761 | 0.9869 | 0.9828 |
| Malconv | 0.8876 | 0.0509 | 0.8999 | 0.8606 | 0.8798 |
| SMO | 0.9035 | 0.0291 | 0.9697 | 0.8430 | 0.9019 |

**Table 4 The time and memory consumption.**

| Model | ProcGCN | Malconv | SMO (on CPU) |
|---|---|---|---|
| Preprocessing time (total/s) | 34,120 | 224 | 3,847 |
| Prediction time (per step/ms) | 3 | 5 | 1 |
| Memory (MB) | 1,883 | 2,793 | 137 |

is still far greater than that of the classical SMO algorithm. In terms of preprocessing speed, ProcGCN needs to run IDA_PRO for decompilation, so it consumes significantly more time than other methods. For memory consumption, ProcGCN model is implemented with sparse matrix, which saves memory consumption. Malconv needs to input the entire byte sequence, which occupies a lot of memory.

In summary, the proposed ProcGCN model has achieved a good detection effect and prediction speed, which proves that ProcGCN can effectively detect malicious processes in memory images.

# CONCLUSION

This article introduces the ProcGCN model, a deep learning model based on the DGCNN model to detect malicious processes in memory images. Firstly, the FCG of the process is extracted from the whole system memory image, the feature vectors for the function node names in the FCG are generated based on the word bag model, and then the FCG is input into the ProcGCN model for classification detection. In the experiment, the ProcGCN model achieved an accuracy of 98.44%, an F1 score of 0.9828, demonstrating obvious advantages in detection effect and speed. These results verify the effectiveness of the method based on graph representation learning and static function call features in memory forensics. In the future, we will further investigate using more memory information as node and graph level features, as well as using the model to detect infected processes.

# ACKNOWLEDGEMENTS

The authors would like to thank all the reviewers who participated in the review, as well as MJEditor for providing English editing services during the preparation of this manuscript.

### Funding
This work was supported by the National Natural Science Foundation of China (No. 60903220). The funders had no role in study design, data collection and analysis, decision to publish, or preparation of the manuscript.

### Grant Disclosures
The following grant information was disclosed by the authors:
National Natural Science Foundation of China: No. 60903220.

### Competing Interests
The authors declare there are no competing interests.

### Author Contributions
- Heyu Zhang conceived and designed the experiments, performed the experiments, analyzed the data, performed the computation work, prepared figures and/or tables, authored or reviewed drafts of the article, and approved the final draft.
- Binglong Li conceived and designed the experiments, performed the experiments, analyzed the data, authored or reviewed drafts of the article, and approved the final draft.
- Shilong Yu performed the experiments, performed the computation work, prepared figures and/or tables, and approved the final draft.
- Chaowen Chang conceived and designed the experiments, analyzed the data, authored or reviewed drafts of the article, and approved the final draft.
- Jinhui Li performed the experiments, performed the computation work, prepared figures and/or tables, and approved the final draft.
- Bohao Yang performed the experiments, performed the computation work, prepared figures and/or tables, and approved the final draft.

### Data Availability
The code and data are available at GitHub and Zenodo:
- https://github.com/zzz4158/procgcn.
- Zhang, H. (2023). The code of ProcGCN. Zenodo. https://doi.org/10.5281/zenodo.10690925.

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
