# Peer review of "ProcGCN: detecting malicious process in memory based on DGCNN"

_PeerJ Computer Science, doi:10.7717/peerj-cs.2193_

## Round 0.1 · original submission · Major Revisions

The authors need to improve the theoretical analysis in order to prove the feasibility of the proposed method. This underpins the research aim, so it is critical that this is addressed in the revision.

·

Basic reporting

no comment

Experimental design

Please add the experimental design in section "4.2 Experimental Environment and Procedures" and explain why the data split is 9:1 instead of the traditional 8:2.

Validity of the findings

In order to verify the validity of the experimental results, please compare the experimental results with the latest research.

Additional comments

This article is less like a scientific study and more like an experiment.

Cite this review as

Reviewer 2 ·

Basic reporting

no comment

Experimental design

The authors mention in the abstract and introduction that byte features are susceptible to mutation and obfuscation, and function call features are more robust. However, there is no experiment to prove this. The evaluation result shows that byte features perform equally well. I'd suggest that the authors evaluate some packed or obfuscated samples to demonstrate the effectiveness of process dumps.

On the other hand, ProcGCN is only evaluated against MalConv and Bozkir's method. Section 2 of the paper mentions other open-source works, such as DMalNet, that utilize similar approaches. I'd suggest that the authors evaluate more related work to show the robustness and effectiveness of ProcGCN.

Validity of the findings

The evaluation of the proposed approach relies solely on the dataset collected by Fang et al. Moreover, the authors randomly selected 2,000 samples from the dataset and retained only the ones whose call graph contains between 20 to 1,000 nodes.

First, this dataset is insufficient for a valid malware detection evaluation. With fewer than 900 malicious samples in the dataset, only about 90 malicious samples are used for evaluation. Even with a reported accuracy of 98% to 99%, the result is not convincing, given the size of the dataset. VirusTotal provides malware samples for academic research for free. There are many other services like VirusSign that provide malware feeds for free, so finding executable malware shouldn't be an issue.

Second, the authors need to justify why they randomly selected 2,000 samples and filtered out samples with more than 1,000 nodes. The original dataset is already small, with less than 8,000 samples, and picking out only one-third of the dataset doesn't seem reasonable.

Additional comments

Some minor issues:

1) I'm curious about the average percentage of external functions in a call graph. Since internal functions do not carry semantic information and each GCN layer can only propagate information to its nearest neighbor, the model may not perform very well if the external functions are not close enough, for example, more than four hops away.

2) The authors mention at line 293 that EXEs and DLLs are extracted from the dataset. How do the authors run DLLs in the Cuckoo sandbox? Is there a specific loader submitted to the sandbox?

3) At line 319, there is no need to convert the GDL format to DOT because IDA Pro is able to dump graphs in the DOT format directly.

Cite this review as

---

## Round 0.2 · Major Revisions

According to the reviewers' suggestions and my opinion, I suggest the paper should make major revisions.

**Language Note:** PeerJ staff have identified that the English language needs to be improved. When you prepare your next revision, please either (i) have a colleague who is proficient in English and familiar with the subject matter review your manuscript, or (ii) contact a professional editing service to review your manuscript. PeerJ can provide language editing services - you can contact us at [email protected] for pricing (be sure to provide your manuscript number and title). – PeerJ Staff

·

Basic reporting

no comment

Experimental design

no comment

Validity of the findings

no comment

Additional comments

In this article "4.3 Analysis and Discussion", there are the following two issues that need to be added to the necessary analysis.

Although the performance advantages of the ProcGCN model in experiments were mentioned, the generalization performance of the model was not discussed in detail. Achieving high performance on a small-scale data set does not necessarily mean that the model will perform well on a larger, more complex data set. Generalization performance is one of the key factors in evaluating the effectiveness of a model.

Although it is mentioned that the accuracy of the model improves with the increase of features, the impact of hardware resource limitations on model performance and resource consumption is not discussed in detail. This is an important factor, especially in practical applications where computing resources and memory resources need to be considered.

Cite this review as

Reviewer 2 ·

Basic reporting

no comment

Experimental design

no comment

Validity of the findings

The authors mention that "many internal functions also carry meaningful names", but internal functions typically have no names unless they are also exported functions (which are quite rare in EXEs) or if the corresponding debug information is somehow loaded by IDA. I would recommend that the authors provide statistics on the EXE-to-DLL ratio and the percentage of functions having no meaningful names for both benign and malicious samples.

Additional comments

I'd like to thank the authors for revising the manuscript and answering many of my questions.

1. For my first point, my suggestion was to feed raw binary files instead of memory dumps to the MalConv model. If the MalConv model does not perform well on obfuscated or packed binaries but ProcGCN is resilient to obfuscation, it proves the superiority of using memory images (to some extent).

2. Since ProcGCN relies on memory images, I do not think this is a pure static approach. Hence, I believe comparing it with some dynamic approaches such as DMalNet is reasonable. DMalNet also runs malware in the Cuckoo sandbox.

That being said, the above suggestions are based on the impression that ProcGCN is a generic malware detection approach. If the authors try to propose an approach designed specifically for memory images, I'm fine with the current experiment setup.

3. In GCN, a graph with 1,000 nodes is a relatively small graph and should not cause any memory stress. I guess the authors are keeping the edges in a dense adjacency matrix. In that case, using a sparse matrix could potentially allow for handling orders of magnitude more nodes.

4. Line 388, the long*er* feature vector of each node, ...

Cite this review as

---

## Round 0.3 · Major Revisions

Please take the comments of the reviewers seriously and make necessary revisions. If the comments of the reviewers cannot be met, the manuscript will be rejected.

·

Basic reporting

no comment

Experimental design

The article mentioned that there are obvious differences in the preprocessing time of the three methods of Malconv, SMO, and ProcGCN, but no accurate experimental analysis was conducted. Please conduct reasonable experimental settings and verify them.

Validity of the findings

There are only a few experimental results and the conclusions of this article cannot be verified. Please add necessary experiments.

Cite this review as

Reviewer 2 ·

Basic reporting

no comment

Experimental design

no comment

Validity of the findings

no comment

Additional comments

I'd like to thank the authors for submitting the revised manuscript. The authors have resolved all my concerns. While it would be nice to have a generalization (on a completely different malware dataset, e.g. VirusTotal, that has not been trained on) and a real-world performance evaluation (false positive rate on large-scale benign programs), I understand that some of these evaluations might not be feasible within the scope of this study, and it doesn't significantly detract from the quality of the work.

Cite this review as

---

## Round 0.4 · accepted · Accept

The paper has addressed all questions.

·

Basic reporting

no comment

Experimental design

no comment

Validity of the findings

no comment

Additional comments

In response to the revision suggestions previously made, this paper has been revised and there are no major problems now.

Cite this review as

Reviewer 2 ·

Basic reporting

no comment

Experimental design

no comment

Validity of the findings

no comment

Additional comments

I'd suggest that the authors report the total time taken to process the entire test set, rather than the time per step. The per-step measurement can be significantly affected by OS context switches, interruptions, CPU-GPU memory transfers, and other factors, especially at the scale of a few milliseconds. Currently the reported prediction time is not particularly meaningful.

Cite this review as